# We All Have a Role to Play: Redressing Inequities for Children Living with CAH and Other Chronic Health Conditions of Childhood in Resource-Poor Settings

**DOI:** 10.3390/ijns6040076

**Published:** 2020-09-25

**Authors:** Kate Armstrong, Alain Benedict Yap, Sioksoan Chan-Cua, Maria E. Craig, Catherine Cole, Vu Chi Dung, Joseph Hansen, Mohsina Ibrahim, Hassana Nadeem, Aman Pulungan, Jamal Raza, Agustini Utari, Paul Ward

**Affiliations:** 1CLAN (Caring & Living As Neighbours), Denistone 2114, Australia; cath@clanchildhealth.org (C.C.); joehansen223@gmail.com (J.H.); 2College of Medicine & Public Health, Flinders University, Adelaide 5042, Australia; paul.ward@flinders.edu.au; 3CAHSAPI (Congenital Adrenal Hyperplasia Support Group of the Philippines), Manila 1000, Philippines; alain@friarminor.com; 4Department of Pediatrics, Philippines General Hospital, Manila 1000, Philippines; sioksoan@gmail.com; 5Institute of Endocrinology and Diabetes, The Children’s Hospital at Westmead/Discipline of Child and Adolescent Health, University of Sydney, Camperdown 2006, Australia; m.craig@unsw.edu.au; 6School of Women’s and Children’s Health, UNSW Medicine, Kensington 2052, Australia; 7Department of Pediatric Endocrinology and Metabolism, National Children’s Hospital, Hanoi 100000, Vietnam; dungvu@nch.org.vn; 8National Institute of Child Health (NICH), Karachi 75510, Pakistan; mohsinaibrahim@yahoo.com (M.I.); drhassana70@yahoo.com (H.N.); drjamalraza@gmail.com (J.R.); 9Department of Child Health University of Indonesia, Dr. Cipto Mangunkusumo Hospital, Jakarta 10430, Indonesia; amanpulungan@mac.com; 10Department of Pediatrics, Faculty of Medicine, Diponegoro University, Semarang 50275, Indonesia; agustiniutari@gmail.com

**Keywords:** congenital adrenal hyperplasia, inequity, community development, human rights, child, parents, community, chronic, non-communicable diseases, poverty

## Abstract

CLAN (Caring and Living as Neighbours) is an Australian-based non-governmental organisation (NGO) committed to equity for children living with chronic health conditions in resource-poor settings. Since 2004, CLAN has collaborated with a broad range of partners across the Asia Pacific region to improve quality of life for children living with congenital adrenal hyperplasia (CAH). This exploratory case study uses the Knowledge to Action (KTA) framework to analyse CLAN’s activities for children living with CAH in the Asia Pacific. The seven stages of the KTA action cycle inform a systematic examination of comprehensive, collaborative, sustained actions to address a complex health challenge. The KTA framework demonstrates the “how” of CLAN’s approach to knowledge creation and exchange, and the centrality of community development to multisectoral collaborative action across a range of conditions, cultures and countries to redressing child health inequities. This includes a commitment to: affordable access to essential medicines and equipment; education, research and advocacy; optimisation of medical management; encouragement of family support groups; efforts to reduce financial burdens; and ethical, transparent program management as critical components of success. Improvements in quality of life and health outcomes are achievable for children living with CAH and other chronic health conditions in resource-poor settings. CLAN’s strategic framework for action offers a model for those committed to #LeaveNoChildBehind.

## 1. Introduction

Congenital adrenal hyperplasia (CAH) is a spectrum of autosomal recessive disorders of adrenal steroidogenesis, caused by an enzyme deficiency in the adrenal cortex (21-hydroxylase in approximately 95% of cases [1]). Reduced activity or absence of 21-hydroxlase impairs cortisol and aldosterone production, and diverts hormone biosynthesis precursors to excess androgen production. Whilst CAH is unquestionably a complex health condition to manage, internationally recognized clinical guidelines now provide evidence-based recommendations regarding the appropriate management of CAH to support affected individuals so they might enjoy the highest quality of life possible [1,2,3].

Despite this depth of knowledge—and the many educational materials, tools and other resources available to facilitate positive health outcomes—for some people living in low- and middle-income countries around the world in 2020 it could be argued that not much has changed since DeCrecchio’s first description of CAH and the life and tragic premature death of Giuseppe Marzo back in 1865 [4]. In low-income countries where there is no newborn screening (NBS) for CAH [5], low prevalence, high mortality and unequal gender distribution (with male infants dying of undiagnosed salt-wasting crises) are more likely rapid quantitative indicators of inequities of access than they are of low incidence [6], with many of the world’s children failing to benefit from the diagnostic and therapeutic solutions for CAH that have been evolving since the 1950s. Glucocorticoid (hydrocortisone, prednisone and dexamethasone) and mineralocorticoid (fludrocortisone) replacement therapies have proven safe and effective in the maintenance and emergency management of CAH for over 60 years, and NBS promotes timely diagnosis, yet neither are routinely nor affordably available globally at the present time. Indeed, inequities are not always exclusive to low- and middle-income countries: despite the clear advantages of NBS for CAH, some high-income countries (such as Australia) have still not implemented universal nationwide NBS programs, and the resultant preventable morbidity and mortality are increasingly difficult to justify [7,8,9,10].

In 1999, the lived experience of almost losing a three-week-old infant to an adrenal crisis caused by undiagnosed CAH led one Australian (author Kate Armstrong) to appreciate first-hand the dichotomous juxtaposition of the potential for extreme and preventable trauma associated with CAH, against the relative ease of achieving an excellent quality of life once a diagnosis was made and access to affordable, quality healthcare made available. In 2004, this newfound appreciation of the potential for CAH to impact differentially on peoples’ lives was further enhanced by a growing awareness of the experiences of children and families living with CAH in Vietnam. Articles in Australian [11] and American [12] CAH support group newsletters spoke of widespread mortality and morbidity, financial devastation, social isolation and stigma associated with CAH in Vietnam, at a time when young people living in neighbouring countries such as Australia and New Zealand were celebrating personal, educational and vocational achievements in lives relatively unscathed by the same health condition. These reports raised important questions. What were the barriers and challenges faced by families in Vietnam, and what actions could be taken to redress the inequities? What might it take for every child living with CAH in Vietnam to enjoy a quality of life on par with that of children in Australia and other high-income countries around the world?

### What Is Caring and Living as Neighbours (CLAN)?

Founded in 2004 in response to a growing awareness of adverse outcomes for children living with CAH in Vietnam, CLAN (Caring and Living as Neighbours) [13] is an Australian non-governmental organisation (NGO) committed to a rights-based approach to optimizing quality of life for children and young people living with CAH and other chronic health conditions in resource-poor settings.

A published survey of 53 families of children living with CAH in Vietnam in 2005 [14] led to the development of CLAN’s rights-based, strategic framework for action (Figure 1) which seeks to focus multisectoral collaborative efforts on five pillars considered essential to achieving the highest quality of life possible for communities of children living with chronic health conditions:Affordable access to essential medicines and equipment;Education, research and advocacy;Optimisation of medical management (with a focus on primary, secondary and tertiary prevention);Encouragement of family support groups; andReducing financial burdens and promoting financial independence.

CLAN’s framework maintains communities of children living with specific chronic health conditions as the visual hub of multisectoral action, and advocates for community development and patient-and family-centred care. Refined over more than 15 years, and since translated to a broad range of other chronic conditions of childhood (including type 1 diabetes, osteogenesis imperfecta (OI), nephrotic syndrome, rheumatic heart disease and epilepsy), and applied across a number of cultures, communities and countries (including Indonesia, the Philippines, Pakistan, India, Kenya, Uganda and Fiji), CLAN’s Strategic Framework for Action has proven a useful guide for engaging with new partners who are seeking to redress inequities for children with special health needs [15,16,17].

From a sustainability and scalability perspective, CLAN’s goal is to share the “how” of implementing our strategic framework and “five pillars” so that others seeking to redress inequities associated with CAH and other chronic conditions of childhood can do so independently of CLAN. The authors acknowledge CAH as a complex health condition, for which comprehensive biomedical knowledge already exists regarding optimal lifelong management. The objective of this paper, therefore, is not to duplicate existing documented clinical and biomedical knowledge on CAH, but rather to describe the novel community development, multi-sectoral collaborative approach CLAN and our many partners and stakeholders have taken with regards CAH (and, later, other conditions) in Vietnam (and, later, other countries), to provide a model based on the knowledge, tools and products that have been developed over time that might be applicable to drive advocacy and action for CAH and other chronic conditions in other countries.

## 2. Materials and Methods

The underlying ontology, epistemology and methodology of this paper is explicitly acknowledged for full transparency.

The ontology of the first author of this paper (Kate Armstrong) is informed by personal circumstance (as the mother of a young person living with CAH in Australia); professional training (as a medical doctor and public health physician); and lived experience (as the founder and president of CLAN). The co-authors bring clinical and bio-medical expertise from Australia (Maria E Craig), Vietnam (Vu Chi Dung), the Philippines (Sioksoan Chan-Cua), Indonesia (Aman Pulungan, Agustini Utari) and Pakistan (Jamal Raza, Mohsina Ibrahim, Hassana Nadeem); as well as CAH Community (Joseph Hansen and Alain Benedict Yap), organisational (Catherine Cole), public health and sociological insights (Paul Ward). The positionality [18,19] of each co-author differs necessarily: as CAH community insiders and outsiders, medical and public health insiders, CLAN insiders, cultural insiders and outsiders, each brings unique power, knowledge and expertise to this analysis, and we acknowledge the many others who have contributed equally generously to collective efforts for CAH communities in the Asia Pacific region to date (please see Acknowledgements section of this paper).

The epistemological paradigm of critical realism [20] has informed the work of CLAN, and the organisation’s underlying optimism that “change is possible” providing we continue to strive for deeper levels of explanation and understanding. The lived experiences of children and families living with chronic health conditions in resource poor settings are manifestations of underlying causal structures and mechanisms that, once understood and addressed, can be influenced and changed for the better [21,22]. Prioritising the voices of people living with chronic conditions (or the parents of children where their voices cannot yet be shared) reflects CLAN’s commitment to valuing felt needs and patient and family-centred approaches [23], and prioritisation of participatory action research (PAR) when implementing CLAN’s framework [24], and aligns with emerging calls to privilege the voices of people living with chronic conditions within the broader non-communicable disease (NCD) discourse [25].

The authors accept the imperfect nature of all information and knowledge presented, sharing it as a genuine attempt to communicate as objectively as possible a brief insight into the complex series of events, outcomes and impact observed over many years, across a number of countries and health conditions of childhood. A commitment to continuous quality improvement (CQI), reflection, strategic planning, monitoring and evaluation has informed CLAN’s work since 2004, and this paper has been written in response to a request from the editors of the International Journal of Newborn Screening (IJNS) to document the work of CLAN in relation to CAH. As such, this paper does not seek to evaluate CLAN’s work comprehensively and objectively (almost impossible given the authors’ insider positionality), but rather describe and explain it.

The methodology of this research paper is that of an exploratory case study. This particular case study research design is often used to answer questions that ask the “why”, “what” and “how” and can be a precursor to more detailed research or study. It is generally accepted that case studies are best suited to a “comprehensive, holistic, and in-depth investigation of a complex issue” [26], and as such, this research design was deemed well suited to an examination of CLAN’s work to improve health outcomes for children living with CAH in the Asia Pacific region over the last 15 years.

This case study uses the widely accepted Knowledge to Action (KTA) framework [27] developed in 2006 (see Figure 2) to inform and structure analysis. The KTA framework has been acknowledged by the World Health Organisation (WHO) as a useful tool for addressing complex problems in health [28] and facilitates a cohesive summary of the cyclical, PAR nature of CLAN’s work over a prolonged period of time. The KTA framework describes a process whereby existing knowledge and insights are augmented by local contextual adaptation, and implementation plans are then informed in a continuous way by existing and emerging insights. In this way, knowledge creation (and ultimately, the products and tools that are developed over time) and knowledge implementation are dynamically inter-linked, both contributing to resolving the “problem” identified as requiring action.

Graham [27] identified seven phases of the action cycle in the KTA framework, and these informed the analysis in this case study by providing an overarching structure to describe how CLAN’s model was first developed and then implemented to redress inequities for children living with CAH (and later other chronic health conditions) in the Asia Pacific region between 2004 and 2020.

The seven stages of the KTA framework action cycle are:Identify the problem—identify, review and select knowledge;Adapt knowledge to local context;Assess barriers to knowledge use;Select, tailor and implement interventions;Monitor knowledge use;Evaluate outcomes;Sustain knowledge use.

In line with accepted methods for exploratory case studies [29], the data and evidence informing this paper draw on publicly available material, including: insights from an initial health needs assessment undertaken with the CAH community of Vietnam in 2005 (results published in 2006 [14], with ethics approval having been granted by collaborating institutions in Australia and Vietnam); published documents (available in peer reviewed and grey literature and referenced accordingly in this article); historical archives (CLAN annual reports, available online [13] and in the National Library of Australia); and conference abstracts.

## 3. Results

Key activities and historical milestones relating to CLAN’s work to support children living with CAH in Vietnam—and, later, in other countries, and with other chronic health conditions—are presented in the context of the seven stages of the KTA framework’s action cycle. The subsequent impact of these activities on knowledge creation (notably knowledge inquiry, synthesis and the development of products and tools) is documented also.

### 3.1. Identify the Problem—Identify, Review and Select Knowledge

Articles and letters in the 2004 CAHSGA (CAH Support Group of Australia) and CARES (CAH Advocacy, Research, Education and Support) Foundation newsletters [30,31] powerfully described some of the inequities experienced by children and families living with CAH in Vietnam.

Michele Konheiser, the Australian mother of a child with CAH travelled to Vietnam with Professor Garry Warne (Paediatric Endocrinologist, Royal Children’s Hospital, Melbourne), and wrote:

“*I heard stories of parents giving their child one tablet every few days as they couldn’t afford daily medication*.”

“*The supply is unreliable. Sometimes they just can’t get the medication. The black market can, therefore, charge whatever it likes*.”

“*One lady… broke down in tears and I found myself in tears as well. She had so much grief inside her*.”

Prof Warne likewise shared:

“*In Vietnam, many parents cannot afford to buy the drugs their children need*.”

“*One bottle of tablets cost the equivalent of 16 bags of rice*.”

“*No 17OHP testing*… Poor surgical outcomes… Several deaths annually*.”

(*Note: a 17OHP blood test measures 17 hydroxyprogesterone and is routinely used in high-income countries for diagnosing and monitoring CAH)

Clearly, the situation described in Vietnam was completely different to that experienced by CAH families in high-income countries at the time—particularly when compared to countries such as Australia, where universal health coverage (UHC—in the form of *Medicare*) [32] ensures essential medicines and quality health care are affordably available to all.

Reflection on the newsletter articles; personal communication with the authors at the time (Michele Konheiser and Garry Warne); personal insight in to the pain of almost losing a child with CAH; and the lived experience of watching a child with CAH who does not receive medicines deteriorate in a relatively short space of time offered (Kate Armstrong) a unique and acute appreciation of the urgent life and death nature of the families’ plight.

Access to medicines for all children living with CAH in Vietnam was clearly the most immediate and life-threatening problem. Whilst ever essential medicines (glucocorticoid and mineralocorticoid tablet replacement, and hydrocortisone for injection during acute illness) were neither registered nor affordably available in Vietnam, many children with CAH in Vietnam were at risk of dying from a salt-wasting crisis, and discussions around quality of life became largely irrelevant until this humanitarian crisis could be resolved. Estimates at the time (based on discussions with doctors in Australia and Vietnam) suggested there were approximately 300 children across the whole of Vietnam alive with CAH (hereafter referred to as the *CAH Community of Vietnam*) who would require urgent humanitarian aid until such time as longer-term, sustainable, in-country solutions could be identified.

Direct contact was made (by Kate Armstrong) with two companies in Australia who produced the hydrocortisone tablets (Alphapharm—now Mylan [33]) and fludrocortisone tablets (Bristol-Myers Squibb [34]) used by the CAH Community of Australia. Both companies generously agreed to donate enough hydrocortisone and fludrocotisone tablets for the entire CAH Community of Vietnam for a three-year period, and a commitment was made, through the founding of CLAN, that distribution of these medicines would be securely coordinated over the short-term (three years), whilst longer-term, sustainable solutions were sought in collaboration with local partners [35,36,37].

### 3.2. Adapt Knowledge to Local Context

Once access to the essential medicines parents needed to keep their children with CAH alive had been secured, there was a need to consider other priorities and needs of the CAH Community of Vietnam. In this regard, the need for more in depth two-way learning rapidly emerged: CLAN was keen to learn more about the challenges and burdens facing CAH families in Vietnam; and health professionals and families in Vietnam were requesting information and knowledge from Australia that would help children living with CAH in Vietnam achieve the same quality of life and health outcomes people living with CAH in high-income countries of the world were enjoying. There was a need to work together to create and exchange knowledge; as well as share existing (or develop novel) tools and resources that could be used to redress inequities.

With processes for exporting donated drugs to Vietnam established (thanks to generous support from executives of the National Hospital of Pediatrics (NHP) in Hanoi), and mutual agreement reached on the need for strong and transparent systems and partnerships to sustainably facilitate safe delivery of medicines over a three-year period, CLAN received a formal invitation to travel to Vietnam in 2005 to assist with distribution of the medicines, and learn more from local authorities and stakeholders about the best ways to engage moving forward. It was agreed that CLAN would have permission to work in collaboration with health professionals at NHP and Royal Children’s Hospital International (RCHI) Melbourne to survey families of children living with CAH and receiving care at NHP (with ethics clearance received from both institutions) so that a shared understanding of the challenges and barriers faced by the CAH Community of Vietnam might inform future collaborative efforts.

There were no existing validated surveys to draw on, so a template was developed de novo by health professionals and families of people living with CAH in Australia and Vietnam. In order to capture a holistic understanding of the challenges families faced, the survey templates included a focus on:demographic profiles;medication use and purchase;routine management of CAH;management of adrenal crises;health and quality of life;specific challenges experienced by girls living with CAH;understanding other burdens or questions families might have.

The survey was translated into Vietnamese by a local paediatric endocrinologist, and distributed to families at a CAH Club meeting sponsored by CLAN at NHP on 10 June 2005. The results of this health needs assessment (HNA) were presented back to NHP and the CAH community of Vietnam, as well as published in the Journal of Pediatric Endocrinology and Metabolism in 2006 [14].

Acknowledging locally contextualised insights from the surveys would not be available until after the June 2005 CAH Club meeting, preparation for the Club (patient support group) meeting included efforts to identify and share existing knowledge, tools and resources considered valuable by children and families living with CAH in Australia. A CLAN CAH Club newsletter and PowerPoint presentations were developed and a young person living with CAH in Australia (CH) attended the meeting as part of CLAN’s team and co-authored the final paper.

Resources translated into Vietnamese for use by families in advance of the 2005 CAH Club Meeting focused on simple messaging, such as how to optimally use the newly available medicines. There was acknowledgement of the fact hydrocortisone and fludrocortisone tablets had to date been rationed by families, and parents would likely need encouragement to use previously scarce medicines as per the internationally recognized CAH treatment guidelines of the time for routine use (two to three time daily dosing, with triple dosing of hydrocortisone on sick days) [38]. Families were informed of the vital importance of seeking an emergency injection of hydrocortisone from the nearest available health centre to manage adrenal crises if oral hydrocortisone was not sufficient on sick days, and CLAN accepted the advice of local practitioners that it would take time before the Australian practice of ensuring all families had an injection kit at home for use in emergencies could be replicated in Vietnam. Indeed, it took approximately five years before this became common practice, with all families receiving injection kits for the first time at the 2008 CAH Club meetings [39].

As time passed, and the CAH community of Vietnam matured and their knowledge of CAH deepened, more complex educational material was translated into Vietnamese. At the July 2011 CAH Club meeting in Hanoi, the Vietnamese version of C. Y. Hsu and S. A. Rivkees’ comprehensive 290-page book *CAH: A Parents’ Guide* was launched, and free copies of the book were given to all families attending the Club meetings (260 families attended in Hanoi, and another 100 families attended Club meetings at Children’s Hospitals 1 and 2 in Ho Chi Minh City) [40,41,42]. Free copies of the book were also made available to all hospital executives and health professionals caring for children with CAH, and all other CAH families who were not able to attend the Club meetings for whatever reason.

### 3.3. Assess Barriers to Knowledge Use

Analysis of responses to the 2005 HNA informed a more detailed understanding of the underlying barriers and challenges experienced by children and families living with CAH in Vietnam at the time.

Families offered insights regarding the impact of:Unaffordable and unreliable access to essential medicines—overwhelmingly identified as the most urgent priority.Poverty—low incomes (particularly for remote and rural families) were exacerbated by the high cost of medicines and ongoing expenditure and loss of income associated with accessing quality care at tertiary and quaternary centres far from home (few families would trust any health professionals outside of NHP).Knowledge and skills gaps—there was an expressed need for education and training on CAH for children, youth, families and health professionals (especially local doctors, who were not considered knowledgeable enough about CAH to manage adrenal crises). Specific queries around genetic counselling and prenatal diagnosis of CAH were also common.Language barriers—despite the availability of information on CAH in English, almost all families could only speak Vietnamese, and online translation was not yet readily available.Isolation and lack of networks—for both individuals and health professionals.Misinformation and myths—were clearly dominant where there was an absence of accurate information.Social stigma, beliefs and attitudes—notably cultural considerations, such as fears for children around future marriage and procreation prospects.Virilisation—particularly surgical and psycho-social concerns for girls living with CAH when access to medicine had been compromised.Health-system challenges for children living with chronic health conditions—such as the complex referral processes, and gaps in existing universal health insurance systems with regards outpatient care for NCDs of childhood.Travel and transportation challenges for those living some distance from NHP.

Acknowledging the complexity of the situation, and aligned with a critical realist view of causation [43], a “But why?” root cause analysis (also described as the five whys approach [44]) of these insights was conducted to better understand the “*causes of the causes*” [45] impacting adversely on families of children living with CAH in Vietnam. For instance, medicine was not affordably available. But why? It was not sold in local pharmacies. But why? It was illegal for pharmacists to sell it. But why? It was not registered in Vietnam. But why? It was not identified as a high priority essential medicine. But why? There were very few children alive with CAH in Vietnam so there was no national focus, and it was not on the WHO Essential Medicines List for children (WHO EMLc) [46] so was not prioritized at an international level. But why? and so on.

This process was continued in some detail, until all of the challenges and barriers had been analysed in depth by CLAN. Emerging from this process was the identification of five key priorities for action (the “five pillars”) which have since informed CLAN’s model (CLAN’s Strategic Framework for Action):

Pillar 1. Access to medicines and equipment—with a focus on solutions that are short-term (e.g., humanitarian aid), medium-term (e.g., registration of drugs nationally) and long-term (inclusion of essential medicines on national insurance lists).

Pillar 2. Access to education, research and advocacy—including translation of educational material; delivery of training; research and advocacy with a view to empowering families, health professionals and the broader community (locally, nationally and internationally).

Pillar 3. Optimisation of medical management—with a holistic, person- and family-centred approach that encompasses primary, secondary and tertiary prevention, and acknowledges the power of multidisciplinary care.

Pillar 4. Encouragement of family support groups—strengthening networks and partnerships to reduce social isolation and empower families, using tertiary and quaternary government/public hospitals as the geographic hub, and families as the visual hub of collaborative action.

Pillar 5. Actions to reduce financial burdens and promote financial independence—encouraging a broad range of initiatives to reduce the financial burden on families and strengthen capacity of future generations to escape poverty (including but not limited to promoting health insurance, attendance at school and vocational training).

Over time, acknowledging the importance of ethical management and processes, CLAN introduced an internal “Pillar 6” that commits the organization to strong governance and accountability mechanisms. To date this has included a commitment to:Incorporation as an NGO with the New South Wales (NSW) Department of Fair Trade;Fundraising certification with the Office of Liquor and Gaming, NSW;Registration with and annual compliance reporting to the Australian Charities and Not-for-Profits Commission (ACNC);Signatory to the Code of Conduct and annual Compliance Self Assessment (CSA) audits of the Australian Council For International Development (ACFID) [47];Annual reports submitted to the National Library of Australia;Tax deductibility status (TDS) and overseas aid gift deductibility status (OAGDS) with the Australian Taxation Office (ATO) and Department of Foreign Affairs and Trade (DFAT);Formal association with the United Nations Department of Public Information for NGOs (UNDPI/NGO);Special Consultative Status with the UN’s Economic and Social Council (ECOSOC); andCommunity of Practice (COP) member status with the World Health Organisation’s Non-Communicable Disease Global Coordinating Mechanism (WHO NCD/GCM).

### 3.4. Select, Tailor and Implement Interventions

In consultation with a range of partners and stakeholders, a collaborative plan for action was developed in light of the 2005 HNA findings. Some of the actions included in this strategic “Plan for CLAN” and actioned over the last 15 years are summarized in Table 1. Not only was this strategic planning document important in terms of promoting transparent communication of priorities and intentions with local authorities, it also facilitated engagement of a broad range of multisectoral stakeholders in collaborative action, enabling a comprehensive, holistic approach to fast tracking change whilst minimising duplication of efforts to redress inequities. Close communication with local health professionals and CAH Club executive members ensured local priorities and real-time feedback informed ongoing activities.

Maintaining the national “community of children living with CAH in Vietnam” as a visual hub was key to promoting a shared, person-centred mission amongst all stakeholders. With children and families central to all discussions, a sustained focus on shared goals (redressing inequities and improving health outcomes) was achieved. With access to essential medicines secured early (Pillar 1), it was possible to focus limited resources and collective action on other pillars. Translation of educational resources and development of specific tools and resources, such as informational videos in Vietnamese and a Club newsletter (Pillar 2); facilitation of professional development opportunities for staff (Pillar 3); and fundraising to support Club meetings (Pillars 4 and 6) were able to occur in parallel. All such actions ultimately helped reduce financial burdens on families (Pillar 5), until such time as real change was achieved through the profound commitments by the Vietnamese government to implement nation-wide policy changes (such as the inclusion of both hydrocortisone and fludrocortisone on the national essential medicines and insurance list; and implementation of NBS for CAH) which have ultimately had the most profoundly transformational impact on the lives of the CAH Community.

### 3.5. Monitor Knowledge Use

In accordance with CLAN’s internal “sixth pillar”, timely monitoring, evaluation and reporting have been central to efforts to drive change for children living with CAH in Vietnam and beyond.

Reflecting CLAN’s commitment to continuous quality improvement (CQI), indicators used to monitor and evaluate the use of knowledge at the individual, health professional and systems levels are numerous (see Table A1) and include measures of:Conceptual knowledge use—this includes changes in levels of knowledge, understanding or attitudes. Examples of indicators used to monitor knowledge use included: the CAH PepTalk Tool [53], developed to evaluate parental knowledge of CAH and its management; numbers of families and health professionals attending Club meetings and training sessions (reflected degree of engagement); nature of questions posed by families at Club meetings (a useful barometer of the general understanding of the community and tool for identifying widely held myths and misunderstandings); engagement of external partners and stakeholders; and requests to CLAN to scale CAH activities to other hospitals (in Vietnam and beyond) and health conditions.Instrumental knowledge use—monitors changes in behavior or practice (and most importantly, changes that translate into improved health outcomes). Examples of indicators used included: availability and registration of drugs (reflecting the broader health system); use of injection kits on sick days at home by families; patient registers tracking incidence, prevalence, mortality and loss to follow-up; use of growth charts (introduced for routine use in outpatient clinics); availability and quality of educational resources in local language for families and health professionals; availability and analysis of 17OHP and renin testing; use of genetic analysis; and establishment of NBS for CAH.Strategic knowledge use—is the manipulation of knowledge to attain specific power or profit goals (sometimes referred to as “research as ammunition”). Examples of indicators included: publication of results and presentations at international conferences; collaborative engagement in civil society networks; engagement with multilaterals and member state governments; participation of media at Club meetings; requests received to translate CLAN’s model to other conditions and countries; and the number and types of communities established internationally.

In 2015 CLAN collaborated with members of the Asia Pacific Pediatric Endocrinology Society (APPES) [54] to undertake the APPES-CLAN Equity (ACE) “Snapshot Survey”, which provided a rapid landscape analysis of the situation for children living with CAH and three other key endocrine conditions (type 1 diabetes, OI and congenital hypothyroidism (CH)) across the Asia Pacific region with regards to many of the indicators listed above. This consultation process helped to rapidly identify vulnerable communities of children and activities that could be realistically and affordably implemented to improve quality of life for those most at risk, and informed the 2016 Tokyo Declaration (formally approved by APPES Council, November 2016), which acknowledges the inequities facing children living with paediatric endocrine conditions in the Asia Pacific region, and endorses a collective commitment to advocacy, action and sustainable change [51,55].

CLAN’s annual reports (from 2005 to the present) routinely shared key success stories according to the five pillars and planned action model, and ensured all stakeholders were appropriately acknowledged and updated not only on their own important contributions, but how these were further complemented by the generosity and expertise of others. In this way, awareness of the contributions and achievements of other stakeholders served to encourage not only CAH Communities, but also allowed all stakeholders to share an understanding of their own particular contributions to a movement that was achieving sustainable, long-term change for children living in vulnerable circumstances.

### 3.6. Evaluate Outcomes—Impact of Using the Knowledge

CLAN is committed to ongoing monitoring and evaluation, as well as reporting back to key stakeholders in a timely manner. In summarising the impact of collaborative efforts to optimize quality of life for children living with CAH in Vietnam and the Asia Pacific region using CLAN’s model, the RE-AIM Framework [56] provides a useful structure, and considers: Reach; Efficacy; Adoption; Implementation and Maintenance.

The *Reach* of CLAN’s model has been substantial. In Vietnam, CLAN was proud to support CAH Club meetings at the four major children’s hospitals in Vietnam over a 10-year period. Engagement of families at these Club meetings grew annually, and within several years quality educational resources (booklets on CAH in Vietnamese) had effectively been shared with almost 100% of children diagnosed with CAH in the country thanks to the strong partnerships established in Hanoi, Ho Chi Minh City and Hue. Essential diagnostic tests were made affordably available in the tertiary hospitals (such as 17OHP and renin testing), and growth charts introduced for routine use in monitoring. Rapid increases in prevalence after 2005 led to the expansion of a pilot NBS trial in 2006, which has since been expanded across Vietnam. Translation of CLAN’s model to type 1 diabetes (2007) and osteogenesis imperfecta (2011) helped raise the profile of paediatric endocrinology as a medical specialty in Vietnam, and the Vietnamese Pediatric Endocrinology Society was established shortly afterwards. Members of the Vietnamese Pediatric Endocrinology Society now actively engage and lead in regional and international professional meetings, and publish on CAH and diabetes in peer review journals [53,57,58,59,60,61].

Beyond Vietnam, CLAN received a series of requests for collaboration from paediatric endocrinologists for CAH communities in the Philippines (2005), Indonesia (2006) and Pakistan (2007), whereupon the same model of community development was replicated in each country, with local adaptations as appropriate. Access to medicines was a universal challenge for each country initially, and the collaborative efforts to apply for hydrocortisone and fludrocortisone tablets to be included in the WHO EMLc in 2008 was an important step in driving change internationally. All four countries have since seen improvements in access to essential medicines for their CAH communities to varying degrees. For some countries local production and availability of low cost medicines has been achieved (such as hydrocortisone tablets in Indonesia and Pakistan), whilst in others (such as Vietnam), national registration and importation from affordable, quality suppliers have been the preferred solution.

The *Efficacy* of collaborative efforts to date is perhaps best reflected in the achievements seen at local levels of and for CAH communities in the different countries CLAN has worked.

#### 3.6.1. Vietnam

The prevalence of CAH has increased nationally, with an estimated increase from 150 children at NHP in Hanoi in 2004 to 1235 cases (325 with genotyping completed) in 2018 [62], representing a 723% increase over 14 years. Vietnam’s national NBS program now includes CAH (together with congenital hypothyroidism, glucose-6-phosphate dehydrogenase deficiency, galactosemia, phenylketonuria and other inborn errors of metabolism) and has determined the incidence of CAH in Vietnam at 1:9008 live births [63]. Successful implementation of NBS for CAH will continue to strengthen CAH gender equity in Vietnam thanks to early diagnosis and increased survival of males with salt wasting CAH. Molecular diagnosis assists with genetic counselling, and is strengthening diagnostic capacity [64]. A recent publication from Vietnam on genetic analysis of 212 people living with CAH reported six novel variants [62].

Quality of life for the CAH Community in Vietnam is a research priority. Within a few years of the 2005 HNA and sustained actions around the five pillars of CLAN, there were anecdotal reports from local health professionals indicating fewer children were presenting in adrenal crisis; likewise, families shared stories at Club meetings of their experiences using hydrocortisone injections at home on sick days to save their children’s lives. Reductions in mortality had an impact on the age profile of the CAH Community: in 2006 only 7% of all children attending NHP for management of CAH were over the age of 13 years [65], but in 2020 detailed follow up on health outcomes for the community are well underway. Formal analysis of quality of life (using the Health-related Quality of Life in Children PedsQL™ 4.0 measurement model [66]) has been completed with 137 children, and a review of clinical health outcomes on a cohort of 81 children at NHP over the age of 10 years includes measures of growth, body mass index, insulin levels, HOMA-IR (homeostatic model assessment of insulin resistance) and metabolic syndrome criteria [63].

A key concern of parents and young people living with CAH shared during CAH Club meetings over the years related to cultural considerations, and in particular, community fears regarding the impact of CAH on potential marriage and child-bearing prospects for young girls later in life. In 2012, CLAN shared a video in Vietnamese language with messaging from an Australian woman living with CAH who had gone on to have children of her own, and this was very well received by the community [67]. In 2020 there are 10 women with CAH treated from childhood at NHP who have now married and had babies of their own [63].

#### 3.6.2. The Philippines

Whilst newborn screening for CAH had been available nationally since 1996, when CLAN was first approached by paediatric endocrinologists from Manila in 2005, neither hydrocortisone nor fludrocortisone tablets were registered or affordably available in country. Short-term humanitarian donations were required for the estimated cohort of 80 children living with CAH in the Philippines at the time (118 children had been diagnosed by NBS as at January 2008) [68], until sustainable local registration and distribution systems were secured.

CLAN attended the inaugural CAH support group meeting for the CAH Community of the Philippines (CAHSAPI) at the Philippines General Hospital’s (PGH) Pediatric Department on 11 December 2005, and was proud to support an initial three year donation of hydrocortisone tablets (again, with generous support from Mylan). This humanitarian aid occurred in parallel to a range of community development initiatives, including the translation of educational resources in to Tagalog and annual face-to-face CAHSAPI meetings. CAHSAPI members continue to actively engage with one another on Facebook (established in December 2010 by Mr Alain Yap, the founding president of CAHSAPI), with strong support from paediatric endocrinology staff at PGH, and currently has approximately 280 members [69]—many of them now teenagers. As at December 2017 the number of children diagnosed with CAH through NBS in the Philippines was 576 (1:18,083 of the 10,415,695 infants screened) [70].

#### 3.6.3. Indonesia

Since CLAN first sent emergency humanitarian supplies of hydrocortisone and fludrocortisone tablets to colleagues in Surabaya (and shortly afterwards, all children living with CAH in the country for a three year period in total) following a request for support in 2006, much has changed for children living with CAH in Indonesia.

In 2007, the inaugural Indonesian CAH support group (IKAHAK) started in Surabaya and in 2008 IKAHAK helped to establish a national CAH community called KAHAKI. Collaborative efforts by CLAN, Indonesian paediatric endocrinologists, IKAHAK and KAHAKI focused on securing longer-term, affordable access to essential medications for the CAH community members. Following the inclusion of hydrocortisone and fludrocortisone tablets in the WHO’s EMLc in 2008 [68], members of the pediatric endocrinology working group of the Indonesian Pediatric Society began working closely with Indofarma, an Indonesian government-owned pharmaceutical company, regarding the possibility of local hydrocortisone production. Indofarma committed to produce hydrocortisone, however significant time was needed to navigate complex internal factors at Indofarma and other bureaucratic issues. In the interim, medication continued to be sourced in various ways, including from other countries such as the Netherlands, Singapore and Australia to fulfil the ongoing needs of children with CAH. Finally, in 2018, hydrocortisone (Genison) was made available in the market. Hydrocortisone by injection (Fartison) was also produced by a local company in the same year, although fludrocortisone tablets are still not manufactured nationally. One of the drug companies is willing to build a factory to produce fludrocortisone in Indonesia, but efforts have been so far hindered by the COVID-19 pandemic.

In the first year (2009) of the official CAH Registry of the Indonesian Pediatric Society, 69 children were registered (56 females (81%) and 13 males (19%)). This number expanded rapidly, with 439 patients (303 females (69%) and 136 males (31%)) registered in 2020, representing a 536% increase over 11 years. The increase in male patients over this period (from 13 in 2009 to 136 in 2020) represents a 946% increase, which was more than double the rate of change for females (there was a 441% increase in females registered; from 56 in 2009, to 303 in 2020) and may reflect increasing awareness amongst physicians in Indonesia of CAH beyond recognition of ambiguous genitalia in females.

Mortality associated with CAH in Indonesia has reduced over the years, but still represents a significant and preventable inequity. Data reported in Central Java, Indonesia in 2016 indicated ten (13%) of the 78 patients registered with CAH in Central Java since 2009 had died due to adrenal crises. Also of concern, only four (5%) of the 78 patients were male, which may indicate adrenal crises were still going undiagnosed and unidentified in this region at that time [71].

Translation of educational resources on CAH into Bahasa Indonesian has been a collective endeavour, and a locally developed video [72] and booklet specifically addressing the challenges facing teenage girls living with CAH in Indonesia [73] have been very powerful. Emergency cards for families to carry, with instructions written in Bahasa Indonesian have also been available to families since 2019. KAHAKI members use WhatsApp as a tool to communicate with each other, and families use this forum to share their experiences living with CAH, as well as practical tips—such as how to access fludrocortisone tablets.

#### 3.6.4. Pakistan

Communication between CLAN and health professionals at the National Institute of Child Health (NICH) in Pakistan commenced in 2007. Urgent humanitarian donations of hydrocortisone and fludrocortisone tablets were arranged for the estimated 80 children living with CAH in Pakistan at the time, with annual CAH Club meetings culminating in the establishment of CLIP (CAH Living In Pakistan). Key achievements with and for CLIP have included: reduced mortality and loss to follow-up (there are currently 334 receiving care for CAH at NICH alone); collaboration on the WHO EMLc application for inclusion of hydrocortisone and fludrocortisone tablets [74,75]; translation of educational resources on CAH into Urdu; paediatric endocrinology training for local health professionals; support for establishment of SPED (the Society of Pediatric Endocrinology and Diabetes in Pakistan) [76]; development of the SPED app for health professionals (currently has 107 registered health professionals, and provides access to a range of materials to assist with optimization of medical management); and establishment of child psychiatry services. A short-term grant from Pfizer Australia [77] in 2016 enabled CLAN to initiate the employment of a community development officer (CDO) at NICH, which continues to this day and has proven an effective model for facilitating activities around the five pillars [78].

In 2020 two particularly encouraging developments have been achieved. Firstly, local production of hydrocortisone tablets (CortiCort 10mg produced by Tabros Pharma) in Pakistan at affordable prices has commenced (Rs: 2.2/tablet; AUD: 0.018/tablet). Secondly, a pilot project funded by CLAN and APPES has trialed the delivery of essential medicines to the homes of 50 CAH community members identified as living in the most vulnerable circumstances (those families living in poverty and/or remote/rural were eligible). This practical step towards improving access to essential medicines for children living in the most disadvantaged circumstances has already proven very effective, with near universal compliance and follow-up achieved with all fifty children throughout the COVID-19 pandemic a remarkable outcome.

Since 2004 *Adoption* of CLAN’s model has occurred in more than 10 countries and has been used to address over 10 different NCDs of childhood, thereby demonstrating its adaptability across a wide variety of conditions, cultures, languages, healthcare and political systems.

CLAN does not actively seek new partners to adopt its model. In the early years, positive messaging around CLAN’s work initially spread by word of mouth, with paediatric endocrinologists in the Asia Pacific region communicating directly with one another through their professional networks (notably APPES) on the changes they were seeing for the children and families they cared for. Individual health professionals would approach CLAN privately asking if the model could be replicated in their own setting.

As awareness of CLAN’s achievements grew (through sharing of success stories, tools and resources on CLAN’s website, publications and conference presentations globally) direct requests for assistance would also come from families of children living with CAH and other chronic health conditions in resource-poor settings. In these instances, CLAN would encourage families and health professionals to connect locally and nationally to collaborate jointly with CLAN and other stakeholders to scale change in systematic and sustainable ways. CLAN always made it clear that our focus was on community development: CLAN does not export medicines to families or health professionals where individual patients are the sole recipients (the only exception to this being situations where only one or two children with a particular condition are known to be alive in a country, as was the case in Fiji for Osteogenesis Imperfecta in 2016).

In this regard, a vitally important step in commencing work with new partners in any country, or for any chronic condition of childhood, is to communicate clearly the foundational principles that inform every aspect of CLAN’s work. The principles inform CLAN’s approach to implementing our strategic framework for action and the five pillars, and include:A holistic view of health—CLAN acknowledges the WHO definition of health [79], with a focus on body, mind and spirit, and an appreciation of the impact the socio-cultural determinants of health (SCDOH) [45] have on health outcomes.Human rights-based approach—acknowledging rights and responsibilities as outlined in the United Nations’ Convention on the Rights of the Child [80].Equity—a commitment to strive for excellence for all, and ensuring the rights of children in high- and low-income countries to the highest quality of life possible are respected, promoted and protected.Community development—all children living with the same chronic health condition in a country are members of a community; these NCD Communities are considered as interconnected and united at the local, regional, national and international level.Community control—people living with chronic conditions are experts and must be consulted at all stages when decisions are made around appropriate approaches and actions to drive change.Person- and family-centred care—acknowledges the pivotal role children, young people and families play in all activities. Indeed, a number of parents of children with chronic health conditions have stayed engaged with CLAN over a decade, and have been champions in their country for change.Sustainable, ethical and transparent approaches to project management—CLAN is committed to the highest standards of accountability and reporting required of NGOs (by ACFID) in Australia and to the United Nations (through UNDPI/NGO and ECOSOC); as a not-for-profit CLAN is committed to sustainable approaches and responsible action in the face of climate change.Multisectoral collaboration and partnerships—are key to sustainability and success.Above all do no harm—is an overarching guiding principle and informs all actions.

To date, key learnings from work with CAH Communities in the Asia Pacific (and later Nigeria in 2012) have since been transferred to support children living with:Type 1 diabetes—Vietnam (2007), Pakistan (2007) and Indonesia (2020);Osteogenesis imperfecta (OI)—Vietnam (2011), Indonesia (2013), Pakistan (2014) and Fiji (2016);Duchenne muscular dystrophy (DMD)—Vietnam (2012);Nephrotic syndrome (NS)—Vietnam (2010);Rheumatic heart disease (RHD)—Kenya (2013);Nodding syndrome and epilepsy—Uganda (2017);Thalassaemia—India (2020);Asthma, cancer, autism and cerebral palsy (amongst others)—collaborative advocacy efforts in multiple countries.

CLAN’s model has been validated across conditions, countries, cultures and communities, achieving comparable outcomes and positive impacts in a diverse range of settings. Co-design and local adaptation have always been a feature of *Implementation* of the model, and integral to the successful outcomes and impact seen. CAH communities in different countries had different starting points, so individual national strategic plans had to be developed each time.

One clear indicator of local ownership is often the names chosen by the communities CLAN has worked with across the Asia Pacific region. For instance, CAH Communities we partner with include: CAHSAPI in the Philippines (the name includes a play on the Tagalog word “kasapi” meaning “being a member or part of”); IKAHAK and KAHAKI (a play on words and sounds using “KAH” for CAH in Bahasa Indonesian); and CLIP in Pakistan (an acronym for *CAH Living in Pakistan*). CLAN’s commitment to community development is strengthened by our sixth pillar’s guidance around ethical and transparent management, and our long-term compliance to the rigorous self-assessment and reporting requirements of full signatories to ACFID’s code of conduct reflects this. Whilst the cost benefits of CLAN’s model have not been formally evaluated by health economists, the engagement and uptake of the model (usually for more than one condition in each country) suggests local authorities see value in any implementation costs.

Sustainability is a foundational principle for CLAN’s model, and the need for *Maintenance* of efforts over the longer-term is acknowledged as fundamental to success. That said, whilst some of CLAN’s original stakeholders and partners are no longer actively engaged, this is not necessarily always a negative outcome. For instance, humanitarian donations of hydrocortisone and fludrocortisone are no longer urgently needed in most of the countries CLAN has worked in due to successful efforts by governments, local health professionals and community members to secure longer term, sustainable localized solutions. By contrast, long-term engagement of other stakeholders over the years speaks to the success of CLAN’s model, and particularly in the case of relationships with the CAH Club executive and leaders, indicates the importance of engaging parents and young people living with CAH and other NCDs, given the chronic, life-long nature of most NCDs.

Social media and technology (notably Facebook, WhatsApp, SMS messaging and websites) have proven a cost-effective vehicle for maintaining connections and raising awareness, and are certainly pivotal to maintaining networks amongst young people and families living with CAH in resource-poor communities of the Asia Pacific region. Introductions between CAH Club Members in Vietnam and other international CAH support group networks (notably, Australia, New Zealand, UK, USA, Philippines, Indonesia and Pakistan) strengthen a sense of international belonging and community, as well as opportunities for sharing updates, information and support. International awareness days (such as Wishbone Day [81] for the OI Community on 6 May each year) have proven powerful platforms for community development activities.

### 3.7. Sustain Knowledge Use

In additional to local action, CLAN has always committed to ensuring community priorities and voices inform international advocacy and efforts to raise awareness of the challenges facing children and young people who are living not just with CAH, but rather NCDs more generally. To this end, CLAN was proud to serve as the inaugural Secretariat of NCD Child [82], with a view to promoting the voices of children and adolescents within the international NCD discourse. Early achievements of the NCD Child movement included acknowledgement of children, young people and a life course approach to NCDs within the 2011 UN High Level Meeting Declaration on NCDs [83]; and acknowledgement at the 2013 World Health Assembly that *“(c)hildren can die from treatable non-communicable diseases, such as rheumatic heart disease, type 1 diabetes, asthma, and leukaemia, if health promotion, disease prevention, and comprehensive care are not provided”* (page 8, para 2 [84]). These seemingly simple statements were the first time chronic conditions of childhood had been acknowledged by member states and the WHO in the context of the global NCD discourse, and now offer a powerful platform for future advocacy and negotiation by NCD community members, health professionals and health system officials.

CLAN has continued to engage as a participant in the WHO’s Global Coordinating Mechanism (GCM) on NCDs since transitioning from the Secretariat role of NCD Child in 2014, and more recently has proudly taken on the role of inaugural Secretariat for IndigenousNCDs [85], a movement committed to privileging First Nations voices on the issue of NCDs. CLAN’s rights-based, community development approach seeks to strengthen the capacity of First Nations peoples to advocate for action that addresses the especial inequitable burdens experienced by their children, young people and other community members living with chronic health conditions, too often in resource poor settings.

A consideration of cost is relevant here, and it is important to note CLAN is a not a large NGO, and in fact when assessed according to ACFID criteria ranks amongst the smallest NGOs in Australia. Relying on volunteers and modest donations and grants, CLAN’s commitment to cost-effective sustainable solutions comes as much from necessity as it does from choice. More than ever, high-impact, low-cost initiatives such as publications, social media and website use, networking and collaboration with like-minded organisations will be key to success and sustainability. Whilst annual support group meetings provide the perfect vehicle for early support and longer term engagement, as access to medicine is secured locally, online platforms are established and locally developed tools, products and resources are utilised, the subsequent baseline knowledge, expertise and connectedness of the CAH community and their local health professionals strengthens, thereby reducing reliance on international partners for support.

## 4. Discussion

### 4.1. Limitations and Challenges

It is important to acknowledge there are limitations to the extent to which persons directly involved in or collaborating with the work of CLAN can objectively evaluate the benefits and merit of activities, much less their outcome and impact. Analysis of CLAN’s model using the KTA framework was an attempt to bring some objectivity to the process of describing the work of CLAN and how we have worked with others to overcome the many challenges and burdens facing children living with CAH and other NCDs in resource-poor settings. The authors note more research in this field is urgently needed. Cohort studies and randomized controlled trials with chronic conditions of childhood such as CAH are extremely limited and rarely population based, therefore the data in this report are based on the KTA framework and CLAN’s model rather than more rigorous study designs. Future research could include comparative studies in countries using CLAN’s framework versus those who are not.

Financing for NCDs globally is a major challenge, and there is increasing recognition of the inequitable burden carried by the poorest peoples living in the poorest countries with NCDs such as CAH [86]. To complicate this further, it is not just the poor who are impoverished by NCDs. CLAN’s experience has been that even relatively wealthy families of children living in countries where quality care is not affordably available risk bankruptcy over time, gradually selling all their assets and ultimately joining the ranks of the “poorest billion” as a result of catastrophic health spends in (too often futile) attempts to save their loved one. With this in mind, it is not surprising that chronic underfunding and limited human and material resources are ubiquitous and sustained challenges facing CLAN and other partners working in this space. Encouragingly, experiences over the last 15 years in multiple countries demonstrate innovative approaches which place NCD communities as the central focus of collaborative action have the potential to scale change at national levels for entire populations of children living with chronic conditions in resource poor settings. Investing in communities brings sustainability, and communities themselves are ideally placed to advise on the best ways to strengthen health systems from the grassroots through to the quaternary levels.

Sadly, despite the best efforts, the reality is some lives cannot be saved. In low- and high-resource settings alike, it is imperative humane processes are in place to support families experiencing the trauma of losing their child in the end stages of a chronic health condition (for example cancer or chronic kidney disease). In such circumstances, clear information, supportive health professionals and quality palliative care are needed to reduce the suffering of children and their families alike, and protect parents from their well-intentioned allocation of limited remaining time and funds to chasing elusive cures and expensive therapies that have no evidence base.

### 4.2. Recommendations

Key recommendations emerging from this case study that are offered to others keen to redress inequities for childhood NCD communities would include:

### 4.3. Affordable Access to Essential Medicines and Equipment Is Pillar 1 for a Reason

Over many years CLAN’s experience has been that humanitarian donations of essential medicines to keep children alive must be the first priority. Much like Maslow’s hierarchy of needs [87], affordable access to essential medicines is the foundation on which all future achievements can be built. Whilst it usually takes at least three years of humanitarian donations to keep children alive until local solutions are established, once medicines are affordably available long-term, families are able to focus on thriving, not just surviving.

With local production of affordable, quality hydrocortisone and fludrocortisone now underway in the Asia Pacific region, and both medicines on the WHO EMLc since 2008, there is no justification for any country of the world to deny children living with CAH their basic human rights to life and health. It must be noted here that this is equally the case for insulin, which, some 100 years after its discovery is still unavailable because it is unaffordable to virtually all children and young people living with Type 1 diabetes in low-, middle- and sometimes even high-income countries, such that WHO is now implementing innovative action to increase access globally [88].

### 4.4. Share the Wheel—Don’t Reinvent It

Many health professionals have reached out to CLAN over the years to ask “how does it work” and “can you help?”. Whilst CLAN’s strategic framework for action speaks clearly to the “who” (NCD community, multisectoral stakeholders) and the “what” (the five pillars) of CLAN’s work, the three dimensional model that emerged from this case study (Figure 3—representing the interplay between CLAN’s Strategic Framework for Action and the KTA framework) will hopefully prove a more useful tool for communicating the “how” (local adaptation, tailored interventions, participatory action research approaches, sharing of products and tools etc.) of CLAN’s model to others in future so that efforts can be taken to scale.

CLAN is committed to freely sharing all products, tools and success stories with others so that achievements can be replicated without the need for CLAN to be directly involved. All products and tools referred to in this document (see Table A2 for more details) are available to share, and CLAN acknowledges the many other NGOs working globally to support children living with NCDs and other chronic health conditions. It has been CLAN’s experience through the work of NCD Child and IndigenousNCDs that there is almost universal willingness of NGOs in this space to generously share with others so that children everywhere might benefit.

### 4.5. Community Is Core to Sustainability

On reflection, it is a remarkable achievement for a small NGO to still be operating internationally more than 15 years after the first flash of inspiration. Although a true appreciation of the inherent risks of CLAN’s model was not immediately apparent when CLAN was founded, it certainly became clear over time that the very nature of CLAN’s vision and mission potentially exposed the organization to inherent risk: a rights-based focus on children living with chronic health conditions in resource-poor settings commits CLAN to working with communities who are amongst the most poor, most inequitably burdened, least powerful and least privileged people in the world.

Ironically however, the reverse outcome has been observed. The inherent strengths, commitment, dedication and resilience of these same communities—those labelled “*poorest of the poor*”—has ultimately underpinned CLAN’s greatest successes over time. Of all the stakeholders CLAN has worked with, it is not infrequently the parents of children living with CAH (and later, once conditions improve in a country, young people themselves living with the chronic health condition [65]) with whom we engage most deeply and for the longest periods of time—most usually in their roles as support group executives. For parents of children (and young people themselves) living with the chronic health condition in question, there is a very strong motivation to engage, learn and do what they can to achieve optimal quality of life. There is no question that parents of children living with chronic health conditions in low- and high-income settings alike are almost universally driven by their profound love for their children and sheer determination to do all they can to help their children survive and thrive. The only difference for families in low and high-income settings is the resources and power they have available to them to drive change.

For this reason, when families in resource-poor settings find themselves supported by caring health professionals and other committed partners and stakeholders to redress inequities, they demonstrate time and again a near limitless capacity to contribute to collaborative efforts in very meaningful and deeply committed ways. It is not unusual for CLAN to have worked with some of the same families for over a decade, and this longevity fundamentally impacts on the sustainability of CLAN’s model. In this regard, the importance of maintaining communities of children as the visual hub of multisectoral action cannot be over-emphasised.

On that note, NBS programmes may offer a timely opportunity to systematically integrate and promote CLAN’s community development approach. The diagnosis of a serious, lifelong condition in a newborn baby is almost invariably a traumatic experience for parents, and particularly in resource-poor settings, systematic and early introductions of these families to the relevant NCD Community networks, tools and resources needed to address the five pillars would impact positively on health outcomes and quality of life for individual children. Moreover, a shared commitment internationally to community networks, growth and well-being within the context of national NBS programmes could be a powerful step towards achieving sustained, collaborative action to address inequities for many childhood NCD communities of the world.

### 4.6. Knowledge Is Power

Whilst knowledge creation, and the development of products and tools are important, the real key to redressing inequities is knowledge exchange. We must tailor knowledge to action and action to knowledge, tools and products if we are to strengthen community health literacy. There is already ample evidence available from a variety of settings (high- and low-income alike) regarding the best ways to optimize quality of life for children living with chronic health conditions such as CAH; the real challenge is to translate and localize this knowledge to action in resource-poor settings so that no child is left behind.

Where language is a barrier to accessing knowledge from other community settings, specific efforts must be made to translate material into locally and culturally appropriate resources [89]. Families and health professionals alike require detailed, “deep” knowledge to effectively manage complex chronic health conditions, so although simple, easy to read resources are satisfactory in the short term, over the longer term more comprehensive educational resources for families and health professionals alike should be a major priority. Raising awareness amongst communities of their children’s basic human rights to health and life [80]—and the responsibilities others in society have to protect and promote same—is a core component of CLAN’s work, and our child-friendly rights flyers (available in a range of languages) have been designed to assist with this [49].

### 4.7. Two-Way Learning Strengthens Us All

For those with the capacity for and interest in action, the rationale for involvement need not be simply altruistic. For stakeholders in low-income countries, the cost-benefit of investing in the lives of children and young people who are living with, and at risk of NCDs, can be enormous. Childhood offers a “golden window” of opportunity in the developmental trajectory of future generations, and simple innovations can pay great dividends. For families and health professionals in high-income countries, there is much to learn from our counterparts in resource poor settings and the potential for, and benefits of, two-way learning must not be understated. For instance, Vietnam is now leading the way in genetic analysis for CAH, and has much to offer children, families and health professionals living in Australia and other high-income countries with their expertise in this field. Colleagues in Pakistan are leading the way in the development of mobile phone applications that can provide critical clinical information to health care professionals and service large numbers of families at low cost; and Pakistani pharmaceutical companies are now producing affordable, quality essential medicines for CAH. The Philippines has world class expertise in NBS that many others can learn from, and Indonesia’s innovative educational resources fill informational gaps [90]. A strengths-based approach to collaborative partnerships and two-way learning pays huge dividends for all.

### 4.8. Prioritise Children and Families Experiencing the Greatest Inequities

Knowing where to focus energies to redress inequities can be a challenge. Whilst CLAN has to date responded to requests for assistance where communities and health professionals self-identify urgent concerns, a more systematic approach to identifying those childhood NCD communities most in need of action should be a global priority. Current United Nations Sustainable Development Goals [91] have a limited focus on preventable morbidity and mortality relating to NCDs in children and young people, and this represents a major opportunity for future knowledge creation and action. Investment in taking the APPES-CLAN Equity (ACE) Snapshot Survey to scale may enable rapid identification of communities most in need of urgent action.

### 4.9. Embrace Imperfection amongst Complexity

Whilst the action needed to redress inequities associated with childhood NCDs and other chronic health conditions is not necessarily expensive, it does take time and commitment from a broad range of stakeholders. Holistic, comprehensive approaches are essential. There is never a single easy quick fix, but this should neither paralyse nor prevent us from making a start—a critical realist perspective encourages us that even an imperfect start is better than nothing.

On that note—a bit like *do-re-mi* in the Sound of Music [92]—the five pillars of CLAN provide a very good place to start. Whilst a locally adapted approach is the gold standard, CLAN’s experiences over multiple conditions, countries and cultures have proved to us again and again that stakeholders can be confident action plans developed around the five pillars will set them on the right path until such time as a deeper understanding of local context is achieved. As the song concludes “*Once you know the notes to sing, you can sing most anything*”, so it is with the five pillars of CLAN when tackling complex chronic health conditions of childhood.

That said, realistic expectations are of course essential. CLAN’s model speaks to continuous, long-term approaches. Change will not happen overnight. At least, not until it does.

By way of example, COVID-19—the disease caused by the severe acute respiratory syndrome coronavirus 2 (SARS-CoV-2)—brings new challenges, particularly as the virus presents the greatest risk to people with existing health conditions [93], such that special efforts must be made to ensure the rights of children and young people living with NCDs in both high- and low-income settings are protected and promoted throughout this pandemic. Fragile essential medicine supply chains, over-burdened health systems and financial devastation of families in resource poor settings present real and especial risk to the families CLAN has worked with over many years. Ongoing commitment to equity and health for all will be essential if hard-earned gains are to be maintained. The World Bank estimates COVID-19 will push between 71 and 100 million people into extreme poverty [94] and initiatives ensuring future vaccines go to those who need it, not just to those who can afford it [95], are especially relevant in the case of children living with chronic health conditions in resource-poor settings. Whilst unexpected challenges can be incorporated within the cyclical approach of the KTA and CLAN frameworks, sustained and proactive steps are required by a range of stakeholders to redress inequities. It is essential to maintain flexibility at all times, such that activities and project plans can be rapidly tailored and adapted in real time to adapt to novel circumstances. The scale of some challenges can seem over-whelming at times, but when we work together, and seek collectively to prioritise those living in the most vulnerable circumstances, it is possible to make a difference.

## 5. Conclusions

This exploratory case study of CLAN’s model using the KTA Framework to inform analysis provides practical insights into the real-life actions that can be taken to tackle a complex health problem such as CAH in resource-poor settings, and positively impact on the lives of entire communities of children and families. Until such time as quality health care and affordable access to medicines are universally available to all, CLAN’s five pillars offer a clear roadmap (complete with a range of products and tools that can be shared) for those committed to redressing the inequities facing children living with CAH and other chronic health conditions in low-income settings. Our many partners have demonstrated time and again that such action is scalable, replicable and sustainable.

Communities and local champions are powerful partners in this process; they must be consulted, empowered and informing collaborative action at all times. Health professionals and health systems play a vitally important role, but it is essential for health professionals to reflect upon the fact that health alone cannot achieve health for all. Whilst the United Nations’ Sustainable Development Goals (SDGs) called for global action to Leave No One Behind [96], CLAN’s strategic framework for action offers a roadmap to guide collaborative action so that together we might #LeaveNoChildBehind.

## Figures and Tables

**Figure 1 IJNS-06-00076-f001:**
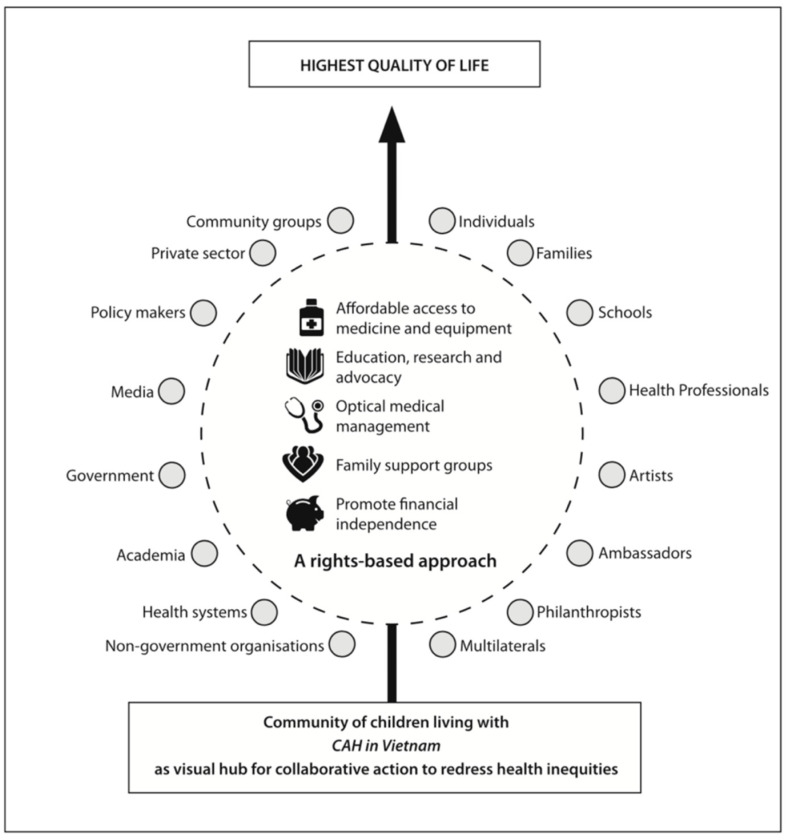
The Caring and Living as Neighbours (CLAN) Strategic Framework for Action (as developed to facilitate change for children living with congenital adrenal hyperplasia (CAH) in Vietnam).

**Figure 2 IJNS-06-00076-f002:**
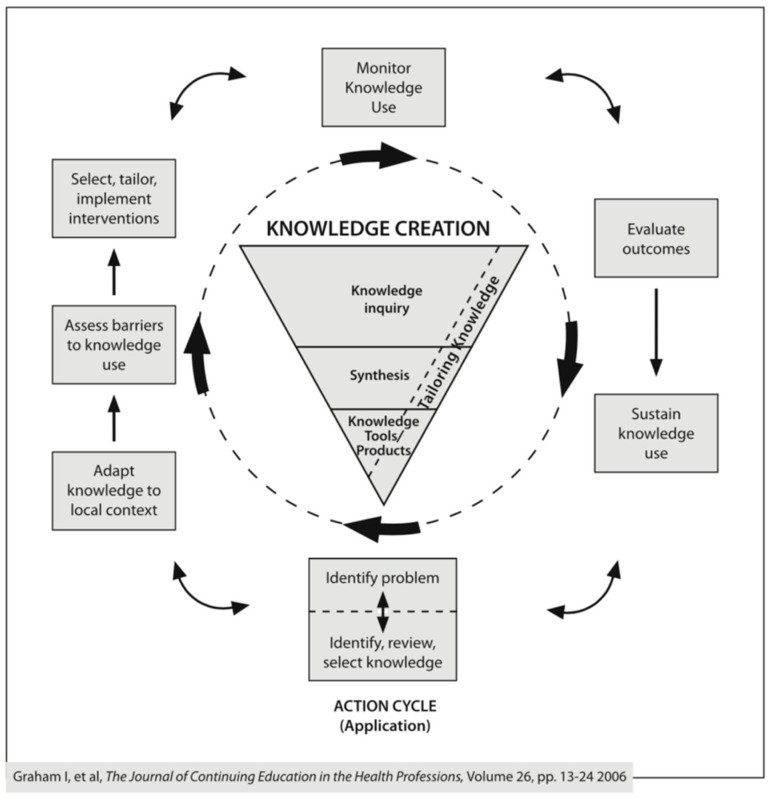
Knowledge to Action (KTA) framework [27] reproduced with permission.

**Figure 3 IJNS-06-00076-f003:**
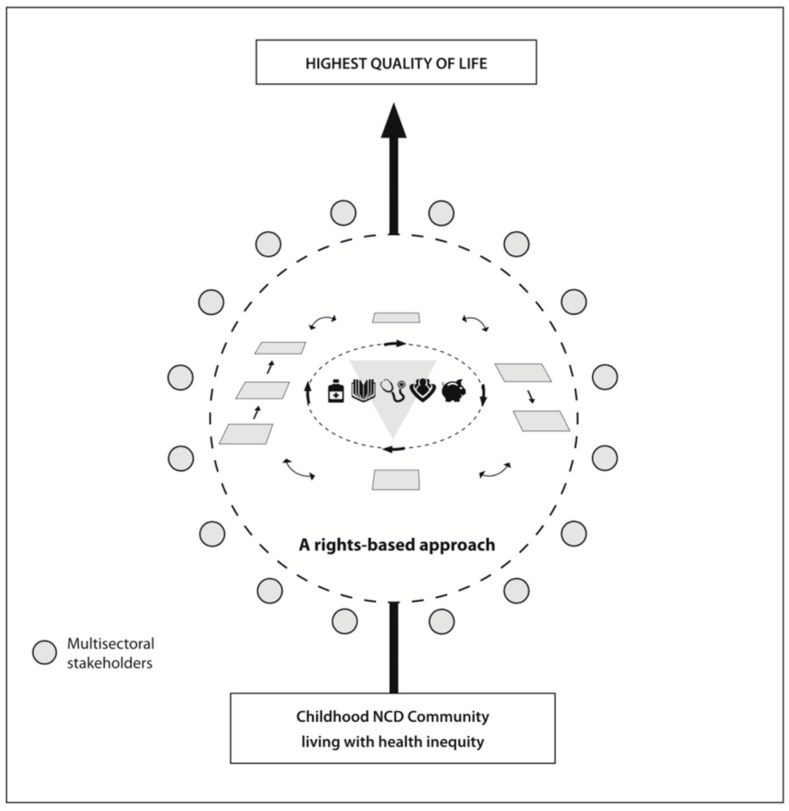
Interplay between CLAN’s Strategic Framework for Action and the KTA framework.

**Table 1 IJNS-06-00076-t001:** Specific examples of tailored, collaborative interventions to support the Congenital Adrenal Hyperplasia (CAH) community in Vietnam.

Priorities	CLAN Activities
Pillar 1. Access to medicines and equipment	*Short-term initiatives* Three year donation of hydrocortisone and fludrocortisone tablets secured; use of hydrocortisone for injection promoted and injection kits shared
	*Medium term* Hydrocortisone and fludrocortisone tablets registered in Vietnam; rapid assessment protocol completed with the International Insulin Foundation [48] to analyse access to medicines in Vietnam; collaborative application to have hydrocortisone and fludrocortisone tablets included in the World Health Organisation Essential Medicines List for Children (WHO EMLc).
	*Long term* Essential medicines for CAH included within national insurance scheme
Pillar 2. Education, research and advocacy	*Education* Translation of educational resources into Vietnamese language; educational sessions for health care professionals (HCPs) prior to Club meetings; educational sessions for families/youth at Club meetings (led by local HCPs); training for HCPs both onsite (Australian nurse educator spent 6 months in Vietnam) and in Australia (endocrinologist training in Australia with APPES (the Asia Pacific Pediatric Endocrinology Society)).
	*Research* Health needs assessment completed and published in journal; RAPIA adapted for CAH and completed in Vietnam [48]; CLAN-APPES Snapshot Survey developed to rapidly identify inequities [15].
	*Advocacy* Presentation on CAH activities at APPES Conference; Child-friendly CAH Rights Flyers (raise awareness of the United Nations (UN) Convention on the Rights of the Child using five pillars) [49]; Club newsletters in Vietnamese (include FAQs; latest information on CAH; messages of support from international community) and videos to raise awareness [50]; Club reports (English) shared with all key partners internationally; success stories/videos shared internationally (CLAN website/social media); CLAN panel at 2010 UN Department of Public Information for Non-Government Organisations (UNDPI/NGO) Conference in Melbourne; APPES Declaration 2018 [51].
Pillar 3. Optimisation of medical management	*Primary prevention* Genetic counselling education and training
	*Secondary prevention* Staff training and education to promote early diagnosis; availability of 17 Hydroxyprogesterone (17OHP) testing for diagnosis and monitoring; newborn screening (NBS) pilot scaled to national program.
	*Tertiary prevention* Staff and family education and training; educational resources and clinical guidelines available in Vietnamese; affordable access to essential monitoring and equipment (such as 17OHP, renin, genetic testing and injection kits); support for gynaecology and surgical teams to exchange internationally [52]; and promotion of growth charts for routine monitoring.
	*Holistic care* Strengthened focus on patient and family centred care; training in psychological support; information about pregnancy for people living with CAH.
Pillar 4. Encourage Support Groups	Support of annual Club meetings; CAH Club executive nominated; communication networks established (Facebook, Whats App, Twitter, Instagram); connections with international CAH support networks facilitated; training sessions facilitated for families and health professionals; success stories from international CAH Communities shared to inspire.
Pillar 5. Reduce financial burdens	Children encouraged to attend school; awards for school performance; education on emergency management/injection kits reduce need to travel; medicine affordably available (on national insurance scheme) and facilitation of international supply chains to optimise pricing; systematic outpatient care (reduce travel and unapproved expenses).
Pillar 6. Ethical & transparent management	CLAN incorporated as non-governmental organisation in Australia [13]; fundraising certification; ethical governance and accountability processes; multilateral engagement, reporting and accountability.

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
