# Peer review of "We All Have a Role to Play: Redressing Inequities for Children Living with CAH and Other Chronic Health Conditions of Childhood in Resource-Poor Settings"

_2409-515X, 2020, doi:10.3390/ijns6040076_

Round 1

Reviewer 1 Report

The authors of "We all have a role to play: Redressing inequities for children living with CAH and other chronic health conditions of childhood in resource poor settings" present a success story of implementing a multifaceted approach in order to adress the inequities in children and families living with CAH  and other chronic health conditions in resource poor settings. The main goal of the NGO CLAN is to adress and improve the quality of life for patients living with chronic conditions where living with one of these disorders, such as CAH, may present with various undue hurdles and problems can be difficult to overcome.

The paper is well written and the approach of CLAN is sound and encouraging for patients and caretakers with other healthconditions other than CAH.

Minor comments:

Figures are somewhat blurry and could stand some improvemt.

Discussion:

In general, the paper reads as a successstory but in the discussion it may be prudent to discuss more problems CLAN's approach may find difficult adressing. What are the limitations?

Author Response

Sincere thanks to Reviewer 1 for these very helpful comments.

The two key points of feedback are addressed as follows:

1) Figures are blurry

  • All three figures have been replaced as EPS files in the document. Separate EPS files will be submitted with the manuscript also (please see attachment).

2) Limitations need to be strengthened

  • A new section on limitations and challenges has been added in the Discussion section. Limitations discussed include: funding and resourcing; the need for more research in this space; poverty; inability to save all children (palliative care and other options are required in some situations); and more details relating to COVID-19 and the challenges this public health crisis brings for children living with chronic conditions in resource poor settings.  
  • Additional details have been included in the section on Vietnam to demonstrate with more practical examples how some limitations were overcome
  • Acknowledgements have also been strengthened, noting all achievements reflect the work of many partners.

Kind regards,

Kate Armstrong

Reviewer 2 Report

The manuscript submitted by Dr. Armstrong and colleagues deals with very practical issues concerning inequities in availability of proper medical care for children suffering from congenital adrenal hyperplasia (CAH) across various Asian countries. This is not a canonical research project, but rather an exploratory case study which implements Knowledge to Action Framework to analyse CLAN’s (an NGO from Autralia) activities for children living with CAH in the Asia Pacific. In a structured and very ordered way the authors present in their paper how a difficult personal experience was coined into precise actions aiming at improvement of the affected children fate in Vietnam. The procedures were then implemented in other low-income countries and with regard to other rare diseases. The grat value of the proposed approach relies upon its comprehensive character, which includes commitment to basic issues, such as affordable access to essential medicines and equipment; education, but also optimization of medical management and above all, encouragement of family support groups. The authors also emphasize ethical, transparent program management and the necessity of respect to the local culture. 

Overall, this is a well-written paper, which presents a complex issues of providing medical care to children suffering from rare disorders in low-income countries. The humanitarian aspect of this story is of great value, and, what is even more inspiring, the practical guidelines are provided. The paper merits publication in International Journal of Neonatal Screening.

Author Response

Sincere thanks to Reviewer 2 for their very generous and encouraging comments. They are very much appreciated.

We are particularly grateful for the acknowledgement of the usefulness of information relating to practical issues and guidelines. We have tried to strengthen this element in the section on Vietnam even further, by including links to videos (such as ref 49) and other practical actions that have been taken to help optimise quality of life for the children and families.

We hope this article is of benefit to others who would like to drive action locally for children living with CAH and other chronic health conditions in resource poor settings, and avoids others having to reinvent as many wheels as possible.

Thank you again.

Kind regards,
Kate Armstrong